# Optimization of Common Iliac Artery Sonography Images via an Indigenous Water Phantom and Taguchi's Analysis: A Feasibility Study

Keng-Yi Wu [1,2,3], Chun-Chieh Liang [1,4,5], Chao-Hsun Chuang [6], Lung-Fa Pan [1,2] and Lung-Kwang Pan [1,*]

1 Department of Medical Imaging and Radiological Science, Central Taiwan University of Science and Technology, Takun, Taichung 406, Taiwan
2 Department of Cardiology, Department of Internal Medicine, Taichung Armed-Forces General Hospital, Taichung 411, Taiwan
3 Division of Cardiology, Department of Internal Medicine, Tri-Service General Hospital, National Defense Medical Center, Taipei 114, Taiwan
4 National Defense Medical Center, Taipei 114, Taiwan
5 Division of Neurosurgery, Department of Surgery, Taichung Armed Forces General Hospital, Taichung 411, Taiwan
6 Department of Pet Healthcare, Yuanpei University of Medical Technology, Hsinchu 300, Taiwan
* Correspondence: lkpan@ctust.edu.tw

**Abstract: Object:** Optimization of common iliac artery sonography images using an indigenous water phantom and Taguchi's analysis was successfully performed to improve the diagnostic accuracy in routine cardiac examination. **Methods:** A water phantom with two major compartments was developed, which satisfied Taguchi's unique criterion of optimization analysis. Two or three levels were assigned to five factors, namely, (A) the probe angle, (B) water depth, (C) sonography preset frame rate, (D) amplitude gain, and (E) imaging compression ratio. The resulting Taguchi's $L_{18}$ orthogonal array contained 18 combinations of 5 factors, ensuring the same confidence level as a realm of 162 ($2^1 \times 3^4$) combinations. The signal-to-noise ratio (S/N) was defined as the minimal difference between the practical survey and predicted areas of 50 mm$^2$ for the sonography imaging scans. The artifact was customized by creating stenosis with a diameter of 8 mm inside a silicon pipe with a diameter of 19 mm. **Results:** The derived optimal parameters included (A) a zero probe angle, (B) water depth of 6 cm, (C) frame rate of 45 Hz, (D) amplitude gain of 50%, and (E) compress ratio of 50% from 3 independent measurements in each group. Further ANOVA confirmed that the frame rate was a dominant factor, with *ss* (sum of squared variances) of 56.6%, whereas the error and other terms were suppressed to 20.3% and 11.9%, respectively. The risks of the inappropriate setting of S/N were also discussed to avoid any misinterpretations. **Conclusions:** The quantified water phantom combined with Taguchi's approach proved to be instrumental in optimizing the sonography image scan quality in routine cardiac examination.

**Keywords:** sonography image quality; common iliac artery stenosis; water phantom; Taguchi's analysis; orthogonal array

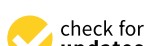



## 1. Introduction

This study aimed to optimize the common iliac artery sonography images and avoid their misinterpretation, thus improving the diagnostic accuracy. The current ultrasound technique is subjective and strongly dependent on operator qualification during the examination's acquisition and interpretation phases. Thus, maintaining high-quality characteristics and reproducibility in routine protocols is challenging in the clinical field. Some limited data for the optimization of sonography images can be provided by available commercial liquid (water, gelatin, etc.) phantoms containing several compartments simulating human body parts [1–4]. In particular, liquid phantoms with a quantified scale are instrumental

in routine quality assurance in clinical surveys. Thus, Grishenkov et al. proposed a customized heart phantom to effectively evaluate myocardial perfusion [5]. King et al. applied a simple liquid phantom to optimize the artifact or uniformity in material and claimed its wide flexibility and low cost [6]. Lo et al. developed a homemade gelatin phantom to simulate the abscess of soft tissues [7]. The water phantom previously proposed by several coauthors of this study provided a clinically confirmed optimization of the left anterior oblique caudal imaging in coronary angiography using Taguchi's method [8]. This study extends the above approach by designing a unique water phantom, which is precisely assembled to simulate real common iliac artery stenosis in clinical examination. Furthermore, every specific factor of the water phantom can be adjusted or replaced to fulfill the researcher's demands. Its application combined with the Taguchi analysis allows one to optimize the performance of sonography imaging.

Most relevant works provide such optimization by a single factor related to one specific quality characteristic. For instance, Zander et al. listed all the possible factors dominating sonographic imaging quality and compared them one by one in optimizing the quality [9], Coffey et al. and Zeng et al. tried to adopt a deep learning-based system to optimize the imaging quality from postprocessing of acquired images [10,11], and Kim et al. claimed that the clutter filtering technique can improve the sensitivity and specificity of power Doppler imaging [12]. In contrast, Taguchi's method, as proposed in this study, provides a robust multifactor analysis on the basis of a conventional setting in routine examination.

The five main factors influencing the sonography image quality are the angle of the probe, water depth, sonography preset frame rate, amplitude gain, and imaging compression ratio. These factors are optimized in this study via the above approach. The assembled water phantom contained two major compartments and properly functioned in controlling the physical characteristics to simulate the bloodstream along the common iliac artery and environmental setting. The analog model representing a clinical syndrome of artifacts in sonography imaging was realized as artificial stenosis with a diameter of 8 mm inside a silicon pipe with a diameter of 19 mm.

Taguchi's methodology and the related calculations were described in the materials and methods, whereas the unique signal-to-noise ratio as a quantified index of quality characteristics was also involved. The dataset from the original sonography multiple scans was processed and discussed in detail. Accordingly, Taguchi's analysis was successfully applied to improve the performance of common iliac artery sonography imaging. The benefit of adopting the acquired area rather than the diameter in the clinical cardiac examination was also analyzed and confirmed.

## 2. Materials and Methods

### 2.1. Study Design

This study was approved by the TAFGH Institutional Review Board committee with credential No. TSGHIRB 2-105-05-089, and the requirement for informed consent was waived.

### 2.2. Taguchi Analysis

Taguchi's analysis is instrumental in optimizing high-quality characteristic systems. This unique method provides special orthogonal arrays to include a large factor's contribution by conducting a limited test series. The derived optimal combination of factors for sonography scan images was tuned off from the environment and other factors. ANOVA (statistical analysis of variance) was performed to evaluate the factors that crucially influenced the target variable. The signal-to-noise (S/N) and ANOVA results were comprehensively compiled to ensure the factor settings for the optimal sonography imaging scan protocol [8,13,14].

### 2.3. Orthogonal Arrays

The sonography protocol settings for the common iliac artery assigned five factors: angle of probe, water depth, sonography preset frame rate, amplitude gain, and imaging compression ratio. Thus, a total set of 162 ($2^1 \times 3^4$) combinations of 5 factors was analyzed (each factor was categorized into 2 or 3 possible levels). The arrangement of samples into only eighteen groups via Taguchi's analysis ensured the same confidence level of results as the conventional optimization of processes [15]. A typical Taguchi $L_{18}$ ($2^1 \times 3^4$) orthogonal array is presented in Table 1, where the digits in each column indicate the levels (i.e., practical arrangements) of particular factors from A to E. Various factors are summarized in Table 2, and the importance of factors is described as follows: (A) angle of probe: the probe needs to be preset vertically or horizontally to suppress the uncertainty in theory while practical surveys always suggest the angle alignment to acquire better echo reflection; (B) water depth: short wavelengths can penetrate a deep water depth and still maintain the imaging quality; yet, under some conditions, short wavelengths imply high frequencies, which cause unexpected noise; (C) sonography preset frame rate: a high frame rate can provide a comparative long persistence of vision, yet, too high a frame rate may also mislead the instant distinguishing of the artifact; (D) amplitude gain (because it can intensify the received signal, but too a high gain may also increase the unwanted noise in reality); and (E) imaging compress ratio: a small compress ratio can provide contrast and sharp image for diagnosis, but too small a setting may ignore essential information and misinterpret the sonography images [16].

**Table 1.** The standard Taguchi's $L_{18}$ ($2^1 \times 3^4$) orthogonal array, where the numbers in each column, except the first column, indicate the specific factor (A–E) level or practical arrangement.

| Group | Factor | | | | |
|---|---|---|---|---|---|
| | **A** | **B** | **C** | **D** | **E** |
| 1 | 1 | 1 | 1 | 1 | 1 |
| 2 | 1 | 1 | 2 | 2 | 2 |
| 3 | 1 | 1 | 3 | 3 | 3 |
| 4 | 1 | 2 | 1 | 1 | 2 |
| 5 | 1 | 2 | 2 | 2 | 3 |
| 6 | 1 | 2 | 3 | 3 | 1 |
| 7 | 1 | 3 | 1 | 2 | 1 |
| 8 | 1 | 3 | 2 | 3 | 2 |
| 9 | 1 | 3 | 3 | 1 | 3 |
| 10 | 2 | 1 | 1 | 3 | 3 |
| 11 | 2 | 1 | 2 | 1 | 1 |
| 12 | 2 | 1 | 3 | 2 | 2 |
| 13 | 2 | 2 | 1 | 2 | 3 |
| 14 | 2 | 2 | 2 | 3 | 1 |
| 15 | 2 | 2 | 3 | 1 | 2 |
| 16 | 2 | 3 | 1 | 3 | 2 |
| 17 | 2 | 3 | 2 | 1 | 3 |
| 18 | 2 | 3 | 3 | 2 | 1 |

**Table 2.** Five factors (each bearing two or three levels) used in the proposed sonography protocol, according to Taguchi's settings in Table 1.

| Factor | Levels | | |
|---|---|---|---|
| | **1** | **2** | **3** |
| (A) angle of probe (degree) | 0 | 30 | |
| (B) water depth (cm) | 5 | 6 | 7 |
| (C) frame rate (/sec) | 45 | 50 | 55 |
| (D) amplitude gain (%) | 50 | 60 | 70 |
| (E) compress ratio (%) | 45 | 50 | 55 |

### 2.4. The Customized Water Phantom

The water phantom was specially customized to fulfill the requirement of multiple factor settings in the Taguchi analysis. The phantom was assembled by two major compartments and accessory components, as depicted in Figure 1A. The left part was the main water tank made from stainless steel, and the top device was an electrical protractor with an adjustable clamp to preset the sonography probe into a demanded angle. The right part was the control box, which had a water pump run by a digital step motor to control the water flow. This phantom was preset to simulate the real blood flow in the common iliac artery, whereas the central electronic device was a noninvasive flow rate meter. This was essential in monitoring the water flow into the silicon pipe for scanning; (B) the inner part of the water tank (L × W × H; 244 × 194 × 202 mm$^3$) included 2 silicon pipes with inner diameters of 19 and 10 mm. In addition, the pipe could be switched alternately by one external valve (the red valve on the bottom of the water tank, refer to 1. (A) and a 19-mm-diameter silicon pipe was also the main surveyed pipe in this work; (C) a 2-mm-thick silicon film 175 mm in diameter, which simulated human skin and could be pressed tightly over the water surface, whereas the water depth could be adjusted by filling or extracting water via the additional drain valve (the blue valve on the bottom of the water tank, refer to 1. (A); (D) a close-up view of the adjustable clamp, which provided the required probe angle for scanning; and (E) the artificial stenosis made by a silicon pipe similar to the pipe adopted in this work, with a diameter of 8 mm and a length of 50 mm. In addition, the stenosis could also be regarded as a block with a tunnel cross-section of 0.5 cm$^2$ (area = $\pi \times 0.4^2$ = 0.50 cm$^2$). Notably, each factor of the phantom could be adjusted independently. Thus, the angle of the probe, water depth, or flow rate could be preset according to Taguchi's suggestion and adjusted to the preset sonography protocol to organize the L$_{18}$ orthogonal array (refer to Tables 1 and 2).

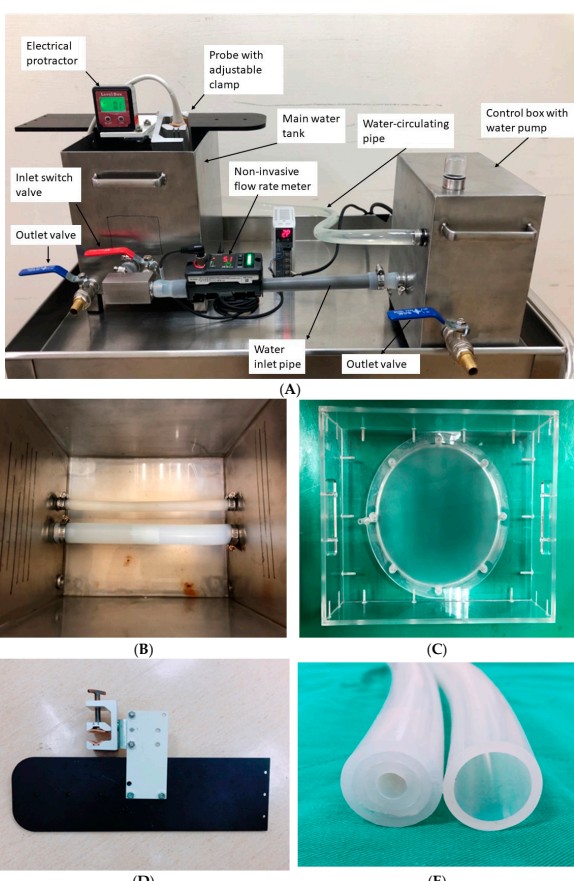

**Figure 1.** (**A**). The left part is the main water tank. The right part is the control box, whereas the central electronic device is a noninvasive flow rate meter. (**B**) The inner part of the water tank

(L × W × H; 244 × 194 × 202 mm³). It includes two silicon pipes with inner diameters of 19 and 10 mm, respectively; (**C**) 2-mm-thick silicon film 175 mm in diameter, simulating human skin, which could be pressed tightly over the water surface; (**D**) a close-up view of the adjustable clamp, which provides the required probe for scanning; and (**E**) the artificial stenosis made by a silicon pipe similar to the pipe adopted in this work.

*2.5. Analysis of Variance (ANOVA)*

A loss function η was defined to capture any deviation between the experimental and desired values, as recommended by Taguchi. The loss function value was transformed into a signal-to-noise (S/N) ratio. The performance characteristics fell into three (lower-is-better, higher-is-better, and nominal-is-best) categories. Each characteristic was associated with a particularly defined S/N ratio used in the computation of the optimal combination of factors. Larger S/N ratios always correspond to better-quality characteristics, regardless of their category. Restated, the optimal level of operating factors had the highest S/N ratio [15]. In addition, the parameters $SS_{Total}$, $SS_{Factor}$, $SS_{error}$, and DoF (degrees of freedom) were defined as follows [17]:

$$SS_{Total} = \left[ \sum_{i=1}^{n} \sum_{j=1}^{r} y_{ij}^2 \right] - n \times r \times \overline{\overline{y}}^2 \tag{1}$$

$$SS_{Factor} = \frac{n \times r}{L} \sum_{k=1}^{L} \left( \overline{y_k} - \overline{\overline{y}} \right)^2 \tag{2}$$

$$SS_{error} = SS_{total} - \sum_{i=1}^{n} SS_{Factor_i} \tag{3}$$

$$DoF_{Total} = n \times r - 1; \; DoF_{Factor} = L - 1; \; DoF_{error} = n \times (r - 1) \tag{4}$$

where $SS_{Total}$ is the sum of squares of all variances; $SS_{Factor}$ is the sum of squares that correlates with the particular operating factor; $SS_{error}$ is the sum of squares of only random errors; $Y_{ij}$ is the specific difference between the theoretical calculation and practical survey of the ith group in the *j*th trial; $\overline{\overline{y}}$ is the average of all derived sonography scan data; $\overline{y_k}$ is the average of the obtained difference that is associated with the specific factor; *L*, *n*, and *r* are the number of assigned levels of the operating factor, number of adopted groups, and number of repeat trials in each group, respectively. The factors are the angle of the probe, water depth, frame rate, amplitude gain, and compress ratio. The corresponding numbers of *L* are equal to 2, 3, 3, 3, and 3, respectively, *n* = 18, and *r* = 3, which ensure reproducibility. $F_{factor}$ is defined as an index in the *F test* to check the specific factor. It can be expressed as in [18]:

$$F_{factor} = \frac{SS_{factor}}{DoF_{factor}} \bigg/ \frac{SS_{error}}{DoF_{error}} \tag{5}$$

where DoF is the number of degrees of freedom. Its values for the angle of the probe, water depth, frame rate, amplitude gain, compress ratio, random error, and total error are 1 [2 − 1 = 1], 2, 2, 2, 2 [3 − 1 = 2], 36 [18 × (3 − 1) = 36], and 54 [(18 × 3) − 1 = 53], respectively. The random error is defined herein as the deviation of the derived data from three independent trials. Thus, the confidence level can be easily derived via the FDIST program run in the Microsoft Excel spreadsheet [19]. The *F* test, which was introduced as early as 1925, is a survey that assumes that the variances of two correlated sample cumulations are equal. If the variances are equal, the probability of the value of *F* exceeding $F_{0.05}$ is only 5% (the latter value depends on the number of samples taken from each cumulation). Thus, $F > F_{0.05}$ implies that the variance of one cumulation is theoretically larger than that of the other. Since $SS_{error}$ is the variance due to the random fluctuation, if factor *A* is expected to influence parameter η, then $F_A$ is most likely to exceed $F_{0.05}$.

*2.6. Sonography Images Scoring*

The artificial stenosis inside the 19-mm-diameter pipe was surveyed three times to ensure reproducibility. Thus, a total of 54 [18 × 3 = 54] trials were scanned and recorded for further analysis. The theoretical diameter or area of the artificial stenosis equaled 0.8 cm or 0.5 cm$^2$, respectively, whereas the obtained data fluctuated in view of various combinations of assigned factors. Either systematic or random error causing the fluctuation in the practical survey can be analyzed according to the signal-to-noise (S/N) ratio, as recommended by Taguchi and listed below. The ideal measured area of artificial stenosis should be 0.5 cm$^2$ (refer to Figure 1E). Thus, the raw data were recorded to assess the difference between the practical measurements and the theoretical value of 0.5 cm$^2$. In contrast, the quality characteristic S/N (unit: dB) was defined as [20]:

$$\frac{S}{N}(\eta_i) = -10\log\left[(Avg(difference)_i \times stdev_i)^2\right] \tag{6}$$

where *stdev* implies the derived standard deviation from three trials of practical measurements in each group. According to Taguchi's recommendation, the sonography image quality characteristic was assessed by the "lower-is-better" principle, and a higher value of $\eta$ was always preferable.

## 3. Results

### 3.1. Raw Data Analysis

The original difference, averages, standard deviations from the original measurements in each group, and S/N values assessed by three independent trials are summarized in Table 3. All values in each group were rearranged per specific factor. The respective three fish-bone plots of the sonography scan protocol are presented in Figure 2. Factor C (frame rate) had the most dominant contribution among all five factors. The highest S/N was revealed at level 1 (45 Hz), although the stdev value was also the highest, namely, 0.04 (refer to Figure 2). According to the results tabulated in Table 3, group 4 outperformed all other groups, with average, stdev, and S/N values of −0.103, 0.035, and 19.72, respectively. The respective factors were as follows: (A) zero probe angle, (B) water depth of 6 cm, (C) frame rate of 45 Hz, (D) amplitude gain of 50%, and (E) compress ratio of 50%.

**Table 3.** The evaluation results obtained from three independent measurements via Equation (6). The area difference is defined as the difference between the practical survey and the theoretical survey (0.5 cm$^2$). The stdev values are the standard deviations obtained in each group from three trials.

| Group | Area Difference (cm$^2$) | | | Ave. | Stdev | S/N |
|---|---|---|---|---|---|---|
| | #1 | #2 | #3 | | | |
| 1 | −0.074 | −0.181 | −0.067 | −0.107 | 0.064 | 19.39 |
| 2 | −0.189 | −0.175 | −0.134 | −0.166 | 0.029 | 15.60 |
| 3 | −0.231 | −0.236 | −0.223 | −0.230 | 0.007 | 12.77 |
| 4 | −0.143 | −0.092 | −0.075 | −0.103 | 0.035 | 19.72 |
| 5 | −0.191 | −0.169 | −0.156 | −0.172 | 0.018 | 15.29 |
| 6 | −0.165 | −0.164 | −0.196 | −0.175 | 0.018 | 15.14 |
| 7 | −0.107 | −0.149 | −0.199 | −0.152 | 0.046 | 16.38 |
| 8 | −0.246 | −0.229 | −0.226 | −0.234 | 0.011 | 12.63 |
| 9 | −0.147 | −0.203 | −0.195 | −0.182 | 0.030 | 14.81 |
| 10 | −0.135 | −0.164 | −0.111 | −0.137 | 0.027 | 17.29 |
| 11 | −0.268 | −0.243 | −0.224 | −0.245 | 0.022 | 12.22 |
| 12 | −0.237 | −0.205 | −0.245 | −0.229 | 0.021 | 12.80 |
| 13 | −0.133 | −0.073 | −0.116 | −0.107 | 0.031 | 19.39 |
| 14 | −0.204 | −0.181 | −0.226 | −0.204 | 0.023 | 13.82 |
| 15 | −0.276 | −0.227 | −0.226 | −0.243 | 0.029 | 12.29 |
| 16 | −0.147 | −0.165 | −0.120 | −0.144 | 0.023 | 16.83 |
| 17 | −0.245 | −0.228 | −0.202 | −0.225 | 0.022 | 12.96 |
| 18 | −0.222 | −0.184 | −0.217 | −0.208 | 0.021 | 13.65 |
| | | | Ave.= | −0.181 | 0.026 | 15.16 |

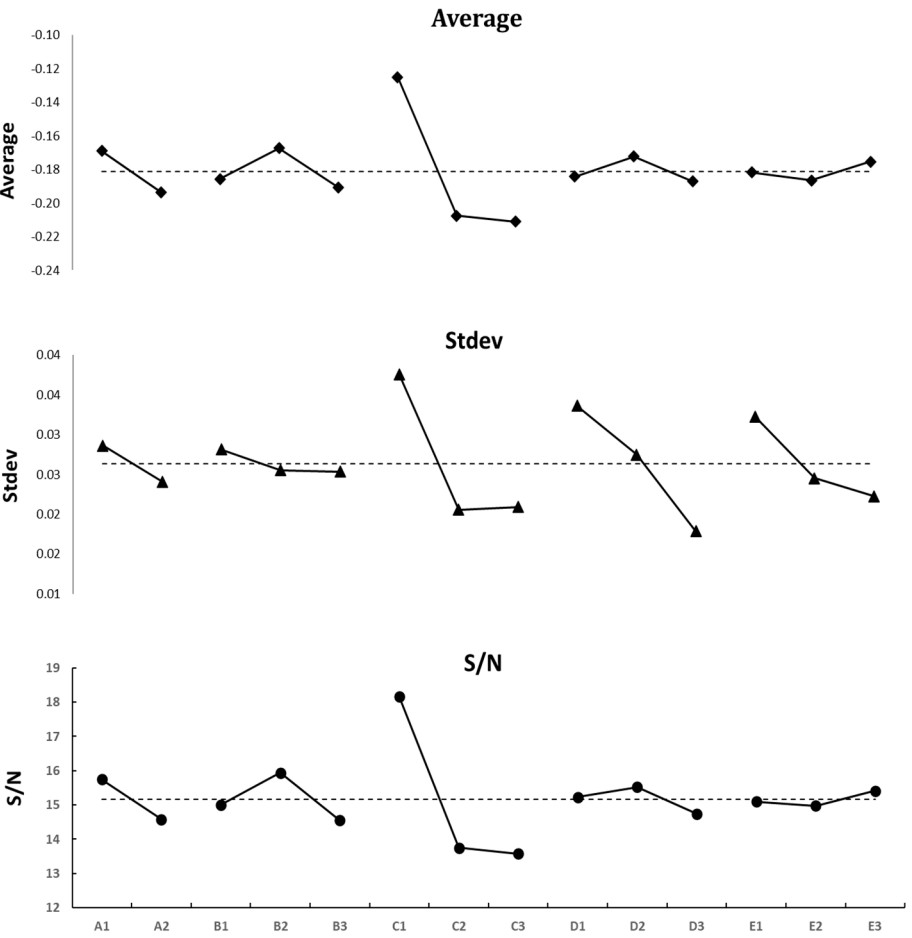

**Figure 2.** The respective three fish-bone plots of the sonography scan protocol. Apparently, factor C (frame rate) has the dominant contribution among all factors.

*3.2. Inspecting the ANOVA*

The sonography scan's dominant factors were double-checked via the F test. Table 4 lists the confidence levels of the factors contributing to the sonography scan integrity. The dominant factors providing the highest contributions to the quality characteristic of the difference between the theoretical calculation and practical survey of 5.4 and 56.6% were A (angle of probe) and C (frame rate), although other factors provided 11.9% of the total contribution. This implies that some intrinsic factors have some influence on sonography performance. In addition, the data fluctuations related to their variation exceeded those of minor factors by 22.3%, according to $SS_{total}$.

**Table 4.** Each factor's confidence level was related to the sonography scan protocol effectiveness. Confidence levels exceeding 99% prove the factor's significance.

| Factor | SS | DOF | Contribution | Var | *F* | Probability | Confidence Level | Significance * |
|--------|-----|-----|-------------|--------|---------|-------------|-----------------|----------------|
| A | 0.0081 | 1 | 5.4% | 0.0081 | 9.5347 | 0.39% | 99.61% | Yes |
| B | 0.0053 | 2 | 3.6% | 0.0027 | 3.1643 | 5.42% | 94.58% | No |
| C | 0.0852 | 2 | 56.6% | 0.0426 | 50.1001 | 0.00% | 100.00% | Yes |
| D | 0.0022 | 2 | 1.5% | 0.0011 | 1.3147 | 28.11% | 71.89% | No |
| E | 0.0011 | 2 | 0.7% | 0.0006 | 0.6502 | 52.80% | 47.20% | No |
| Others | 0.0178 | 8 | 11.9% | 0.0022 | 2.6245 | 2.25% | 97.75% | No |
| Error | 0.0306 | 36 | 20.3% | 0.0009 | | | S = 0.029174761 | |
| Total | 0.1506 | 53 | 100.0% | | | | * Note: At least 99% confidence level | |

## 4. Discussion

### 4.1. Taguchi's Approach Verification

According to Taguchi's original suggestion, the highest S/N in each factor was regarded as the major criterion for the optimal setting of five factors, which ensured the smallest difference between sonography image scans. Despite the high reputation of Taguchi's analysis concerning its wide applicability, its accuracy as applied to this new task needs to be verified. The optimal factors were assigned as follows: (A) 0 degrees, (B) a 6-cm water depth, (C) a 45 frame rate/s, (D) a 60% amplitude gain, and (E) a 55% compress ratio, according to Table 2 and Figure 2. The derived differences between a practical survey and a theoretical survey of 0.5 cm$^2$ from 3 independent surveys of sonography scans were −0.179, −0.092, and −0.171. Thus, the derived values of Ave, stdev, and S/N were −0.147, 0.048, and 16.63 dB, respectively, failing to pass the practical verification. Thus far, Taguchi's theoretical recommendation did not automatically provide the optimal solution unless it passed the practical verification. Therefore, the fourth group turned out to be the optimal group, as assigned in this work. This situation can be attributed to strong cross-interaction (coupling) among factors that biased their contributions. Thus, practical verification was mandatory to suppress the misinterpretation of the derived results. Table 5 lists the derived S/N values for the conventional (first group), a combination of the highest S/N values of each factor, and the optimal (fourth group) by the sonography scanning results. As seen in Table 5, the fourth group had the highest S/N, with high Ave and low stdev values.

**Table 5.** The S/N ratio derived for the conventional (1st group), combined with the highest S/N of each factor, and the optimal setting (4th group) by sonography scanning.

| Factor | Conventional (The 1st Group) | Combined with The Highest S/N | Optimal Setting (The 4th Group) |
|---|---|---|---|
| (A) angle of probe (degree) | 0 | 0 | 0 |
| (B) water depth (cm) | 5 | 6 | 6 |
| (C) frame rate (/s) | 45 | 45 | 45 |
| (D) amplitude gain (%) | 50 | 60 | 50 |
| (E) compress ratio (%) | 45 | 55 | 50 |
| **S/N (dB)** | **19.39** | **16.63** | **19.72** |

### 4.2. Manipulating the S/N Ratio

The original definition of the signal-to-noise ratio (S/N), originally proposed by Taguchi in optimization analysis [15], integrates the practical expectation value and standard deviation in many alternative equations to satisfy various users' criteria [21–23]. However, the S/N definition should satisfy Taguchi's optimization principle, i.e., (1) to minimize the random error first, and then, the systematic error; (2) the systematic error can be effectively suppressed by identifying the adjustment factor, which can change the average of practical data without influencing the random error in measurements. However, a simplified S/N can be given for rapid calculation as adopted by some researchers [14]:

$$\frac{S}{N}(\eta_i) = -10\log\left[(Avg(difference)_i)^2\right] \tag{7}$$

Notably, Equation (7) involves only the difference between the practical measurement and the theoretical value of the derived area from the sonography scanned image by the "low-the-better" principle. In contrast, the proposed Equation (6) emphasizes either a small difference or a low statistical deviation among three independent trials. Accordingly, the derived simplified S/N values of each group were 43.28, 46.48, 56.43, 48.74, 50.33, 49.94, 43.12, 51.97, 45.19, 48.81, 45.34, 46.29, 49.58, 46.78, 43.17, 49.73, 46.24, and 47.36, respectively. The fish-bone plot of the simplified S/N is depicted in Figure 3. As clearly illustrated, the predominant factors became the (D) amplitude gain or (E) imaging compression ratio while the original S/N definition via Equation (6) implied that the dominant factor was the (C) frame rate. However, the frame rate is the most dominant factor from the ANOVA

test ($SS_c$ = 56.6%, refer to Table 4). However, neither the amplitude nor the compression ratio passed the ANOVA inspection (i.e., they were insignificant in the F test). Thus, it is appropriate to emphasize the deviation among repeat trials in each preset group (cf. Equation (6)). In contrast, emphasizing only the difference between the practical survey and theoretical preset to obtain a high S/N ratio (cf. Equation (7)) is not a good option for robust designation, although it provides rapid results for reference. A comprehensive definition of S/N should always integrate a low random error (i.e., *stdev* in this study) and high expectation value (i.e., *Avg*(*difference*) in this study) altogether in the practical survey.

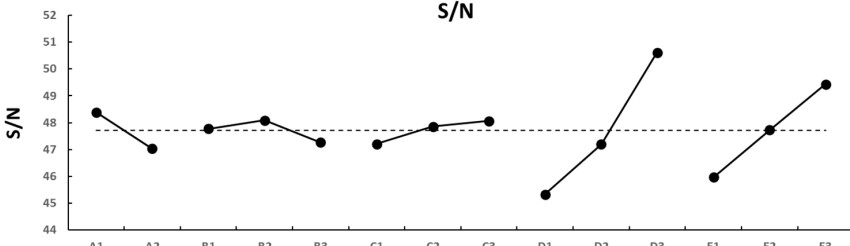

**Figure 3.** The fish-bone plot of the simplified S/N (cf. Equation (7)). The predominant factors become (D) the amplitude gain or (E) imaging compression ratio, unlike (C) the frame rate dominance, according to Equation (6).

### 4.3. The S/N Focused on the Difference in Area or Diameter

In the optimization process, the scoring of the imaging quality in various groups is calculated according to the area difference between a practical survey and theoretical preset. However, alternatively, it can focus on the difference in diameters. In doing so, the data were rearranged in three trials, and the corresponding results are listed in Table 6. The fish-bone plot is depicted in Figure 4. A significant difference between the area- and diameter-based results can be observed. Based on the diameter difference, only factor D (amplitude gain) performs as a significant factor (24.22%) while the contribution of other factors is 42.72% and the random error is 20.61%, indicating a high uncertainty (42.72 + 20.61 = 63.33%). In contrast, based on the area difference, the same derived value is 32.20%, that of the others is 11.86%, and the random error is 20.34% (refer to Table 4). In addition, a lower contribution from the others or random error also implies that a larger contribution comes from those assigned factors and suppresses the harmful interference of other factors. Therefore, focusing on the area difference is more effective than focusing on the diameter difference for solving this particular problem.

**Table 6.** The evaluation results obtained from three independent measurements via Equation (6). The diameter difference is defined as the difference between the practical survey and the theoretical survey (0.8 cm).

| Group | Diameter Difference (cm) | | | Ave. | Stdev | S/N |
|---|---|---|---|---|---|---|
| | #1 | #2 | #3 | | | |
| 1 | −0.238 | −0.238 | −0.195 | −0.224 | 0.02 | 13.01 |
| 2 | −0.092 | −0.082 | −0.044 | −0.073 | 0.03 | 22.77 |
| 3 | −0.072 | −0.039 | −0.039 | −0.050 | 0.02 | 26.02 |
| 4 | −0.022 | −0.022 | 0.021 | −0.008 | 0.02 | 42.31 |
| 5 | −0.044 | −0.119 | −0.119 | −.094 | 0.04 | 20.54 |
| 6 | −0.006 | 0.027 | −0.006 | 0.005 | 0.02 | 46.02 |
| 7 | 0.021 | −0.065 | −0.022 | −0.022 | 0.04 | 33.15 |
| 8 | −0.006 | −0.044 | −0.006 | −0.019 | 0.02 | 34.58 |
| 9 | −0.072 | −0.138 | −0.138 | −0.116 | 0.04 | 18.71 |
| 10 | −0.022 | −0.065 | −0.065 | −0.051 | 0.02 | 25.91 |
| 11 | −0.044 | −0.082 | −0.044 | −0.057 | 0.02 | 24.93 |
| 12 | −0.072 | −0.039 | −0.105 | −0.072 | 0.03 | 22.85 |
| 13 | −0.022 | −0.109 | 0.021 | −0.037 | 0.07 | 28.71 |
| 14 | −0.006 | −0.119 | −0.044 | −0.056 | 0.06 | 24.98 |
| 15 | −0.204 | −0.138 | −0.171 | −0.171 | 0.03 | 15.34 |
| 16 | 0.064 | 0.021 | −0.022 | 0.021 | 0.04 | 33.56 |
| 17 | −0.119 | −0.006 | −0.044 | −0.056 | 0.06 | 24.98 |
| 18 | −0.138 | −0.105 | −0.105 | −0.116 | 0.02 | 18.71 |
| | | | Ave.= | −0.066 | 0.03 | 26.51 |

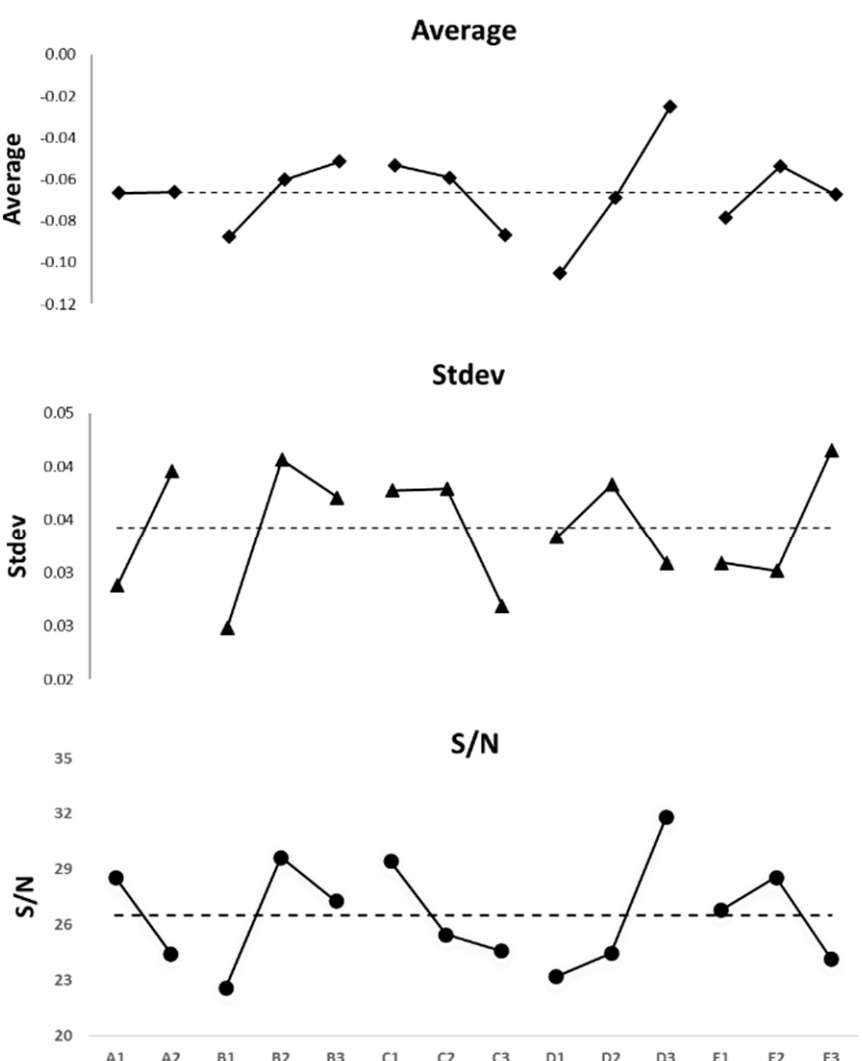

**Figure 4.** The fish-bone plot according to the diameter difference between the theoretical preset and practical measurement. A significant difference between the area- and diameter-based results shows a fluctuating outcome according to two difference scenarios.

*4.4. Clinical Testification*

The optimal preset sonography imaging scan was then clinically tested in the Taichung Armed Forces General Hospital (TAFGH), Taiwan. As shown in Figure 5, two cases were analyzed in this study. Case 1 concerned a female aged 65 who had a diabetic history of 20 years. Her right leg underwent skin flap surgery after a car accident, but the wound did not heal. Computed tomography angiography revealed multiple arterial occlusions in both lower limbs. The CT scanned area (A) was 47.9 mm$^2$, whereas the conventional (C) and optimal (E) settings of sonography scans were 36.8 and 42.5 mm$^2$, respectively. Case 2 included a female aged 88 with a diabetic history of 23 years. She accidentally had an open fracture of the right lower limb. The wound did not heal, and several toes were blackened after emergency orthopedic surgery. A computed tomography angiogram showed multiple chronic arterial occlusions in the patient's right lower limb. The CT scanned area (B) was 17.4 mm$^2$, whereas the conventional (D) and optimal (F) settings of the sonography scans were 24.1 and 20.7 mm$^2$, respectively. In both cases, a better sonography image quality and more accurate derivation were obtained in the optimal scan.

**Case 1** **Case 2**

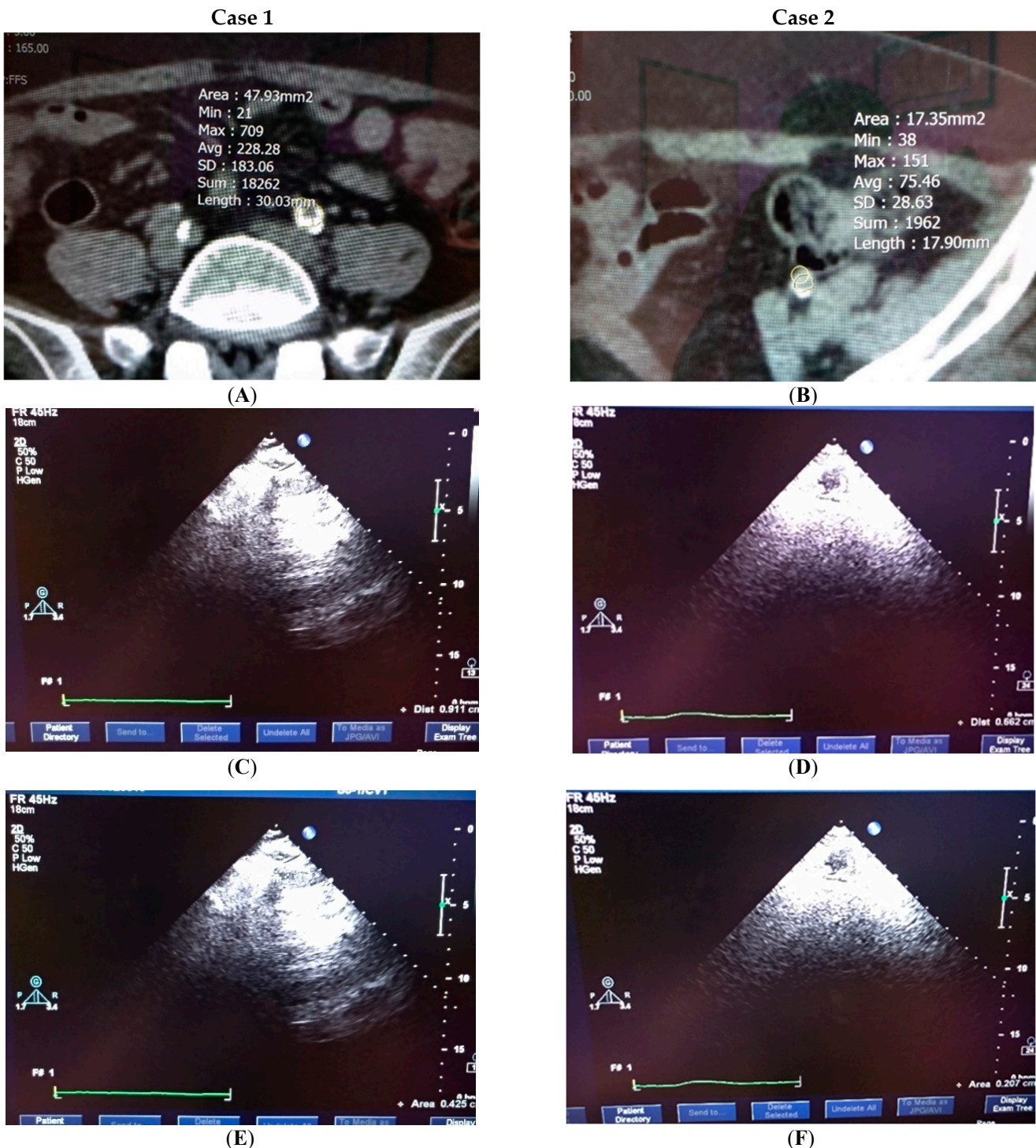

**Figure 5.** Two cases were reported. Case 1: (**A**) CT scanned area of 47.9 mm$^2$, (**C**) conventional area of 36.8 mm$^2$, and (**E**) optimal setting of the sonography scan featuring 42.5 mm$^2$. Case 2: (**B**) CT scanned area of 17.4 mm$^2$, (**D**) conventional setting of 24.1 mm$^2$, and (**F**) the optimal setting of the sonography scan featuring 20.7 mm$^2$.

## 5. Conclusions

The common iliac artery sonography was optimized using an indigenous water phantom and Taguchi's analysis. The water phantom was customized to satisfy Taguchi's criterion of optimization analysis. Accordingly, 5 factors were assigned either 2 or 3 levels to organize an orthogonal array with 18 various combinations of sonography image scan factors to optimize the imaging quality. The signal-to-noise (S/N) ratio was defined to

have a minimal difference in the derived area between the practical survey and theoretical preset at 0.5 cm$^2$. The obtained results were subjected to ANOVA and verified through practical measurements to ensure reproducibility and consistency. The benefit of adopting the area difference instead of the diameter one in sonography scan comparison was proven while alternative S/N definitions were shown to mislead the interpretation of the optimal results. The quantified water phantom combined with Taguchi's approach proved to be instrumental in optimizing the sonography image scan quality in routine cardiac examination.

**Author Contributions:** Conceptualization, L.-F.P. and L.-K.P.; methodology, K.-Y.W.; software, C.-C.L.; validation, C.-C.L., C.-H.C. and K.-Y.W.; formal analysis, L.-F.P.; investigation, C.-C.L.; resources, K.-Y.W.; data curation, C.-H.C.; writing—original draft preparation, L.-F.P.; writing—review and editing, L.-K.P.; visualization, C.-C.L.; supervision, L.-K.P.; project administration, C.-C.L.; funding acquisition, L.-F.P. All authors have read and agreed to the published version of the manuscript.

**Funding:** The authors highly appreciate the financial support of this study by the Taichung Armed Forces General Hospital in Taiwan (contract No. TCAFGH-D-110015).

**Institutional Review Board Statement:** The study was approved by the TAFGH Institutional Review Board committee with credential No. TSGHIRB 2-105-05-089, and the requirement for informed consent was waived.

**Informed Consent Statement:** Informed consent was obtained from all subjects involved in the study.

**Data Availability Statement:** All the correlated data are reported in this article, it is free to download already.

**Conflicts of Interest:** The authors declare no conflict of interest.

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
