# Peer review of "Optimization of Common Iliac Artery Sonography Images via an Indigenous Water Phantom and Taguchi’s Analysis: A Feasibility Study"

_applsci, doi:10.3390/app12168197_

Round 1

Reviewer 1 Report

Authors presented the method Optimization of coronary artery sonography images using the Taguchi analysis of indigenous water phantom. I have major concerns about the manuscript:

Abstract: Add existing problems before explaining the method in the abstract section.

Introduction:

This section is too short and difficult to understand.

There is no motivation of this study, add in the introduction section.

Add your contributions.

Paper structure paragraph is missing.

Related work:

Add separate section for related work which must include the latest works.

Results:

Dataset details are missing needs to elaborate.

Evaluation metrics must be added.

Add comparative analysis with other latest methods.

Reviewer 2 Report

The authors attempted to optimize coronary artery sonography images using an indigenous water phantom. The model is never considered as coronary artery because of the diameter and imaging depth are far from those of coronary artery. Example CT images are not coronary artery. Cross-sectional image of coronary artery is not included in the routine clinical echocardiography.

The theoretical part of the study including Taguchi’s analysis is adequate. The authors are recommended to change the title and the purpose of the study.

Reviewer 3 Report

In the manuscript, entitled 'Optimization of coronary artery sonography images using the Taguchi analysis of indigenous water phantom' by Wu et al., the authors explain to optimize the coronary artery sonography imaging using indigenous water pahtom and Tanughi analysis. 

Nevertheless, the manuscript presents several critical points:

1. The abstract it’s not understandable, it’s confusing. Please reformulate this part.

2. The novelty of the proposed work does not emerge in the article and in the conclusions.

3. The introduction should be better argued, giving valid references to issues relating to high-quality imaging in sonography, the importance of the water phantoms, Taguchi method,  the research design and the novelty of your work, justifying your choises.

4. the references should be improved.

5. When the authors introduce the Tacughi method, they don't give references and appropriate info in order to understand the methodological approach.

6. In my opinion, only two clinical cases can not  give a confirmation. Please, explain this point.

Finally, I would suggest you to carefully review the text.

Round 2

Reviewer 1 Report

Authors done revision very well.

Author Response

thanks for your fruitful comments

Reviewer 2 Report

The authors changed the title from “coronary artery” to “iliac artery” according to one of the reviewers’ comment. However, the details of the ultrasound measurement are still unclear. The image quality of clinical CT and US images is poor. Thus the results of area measurement don’t support the authors’ algorithm.

Author Response

We did our best to mitigate the above three serious criticisms.

  1. Statistical significance is a hard issue to prove even by twice more cases, which could be accumulated only in the next several months. Alternatively, we revised the title and marked this paper as a feasibility study to emphasize that it is pioneering research applying the Taguchi methodology to sonography, and there are still some limitations to further revision in the follow-up research. Our previous success in applying it to other medical examination fields and robust statistical tools have been proven by our previous publications. The accumulated credibility of this methodology and our research team reputation may help to mitigate the raised doubts.
  2. We have revised the quality of Figure 5 from the original CT or sono-scanned images to clearly demonstrate the scanned images from either the conventional or optimal setting of sono images and accompanying CT scans. The revised figure is listed below. Plus, in the subsection “clinical testification” (compared to original “confirmation” with more aggressive term) of discussion,
  3. We quantified the solid proof of derived sono images compared to CT scans as Case 1: (A) CT scanned area of 47.9 mm2, (C) conventional area of 36.8 mm2, and (E) optimal setting of sonography scan featuring 42.5 mm2. Case 2: (B) CT scanned area of 17.4 mm2, (D) conventional setting of 24.1 mm2, and (F) the optimal setting of sonography scan featuring 20.7 mm2. Apparently, the optimal setting can derive more accurate suggestions than the conventional setting in these two cases.